# The Relationship of Cholesterol Responses to Mitochondrial Dysfunction and Lung Inflammation in Chronic Obstructive Pulmonary Disease

**DOI:** 10.3390/medicina59020253

**Published:** 2023-01-28

**Authors:** Bakr Jundi, Huma Ahmed, Joshua Reece, Patrick Geraghty

**Affiliations:** Department of Medicine, State University of New York Downstate Health Sciences University, New York, NY 11203, USA

**Keywords:** cholesterol, cigarette smoke, COPD, inflammation, mitochondria

## Abstract

Hyperlipidemia is frequently reported in chronic obstructive pulmonary disease (COPD) patients and is linked to the progression of the disease and its comorbidities. Hypercholesterolemia leads to cholesterol accumulation in many cell types, especially immune cells, and some recent studies suggest that cholesterol impacts lung epithelial cells’ inflammatory responses and mitochondrial responses. Several studies also indicate that targeting cholesterol responses with either statins or liver X receptor (LXR) agonists may be plausible means of improving pulmonary outcomes. Equally, cholesterol metabolism and signaling are linked to mitochondrial dysfunction and inflammation attributed to COPD progression. Here, we review the current literature focusing on the impact of cigarette smoke on cholesterol levels, cholesterol efflux, and the influence of cholesterol on immune and mitochondrial responses within the lungs.

## 1. Introduction to Hypercholesterolemia in COPD

Reverse lipid transport plays an important role in pulmonary cells, especially alveolar macrophages and type II pneumocytes. Hyperlipidemia, a condition with high levels of lipids in the blood, is common in chronic obstructive pulmonary disease (COPD) patients but its relationship to COPD and lung function is not well established [1]. COPD is the fourth leading cause of death in the US [2]. Cholesterol levels are an important prognostic marker of cardiovascular risk in patients with COPD [3]. Exposure to cigarette smoke is the primary environmental factor associated with the development of COPD. Alpha-1 antitrypsin deficiency is the most common genetic factor linked to COPD. Almost 20% of all coronary heart disease deaths are associated with smoking [2]. Several clinical studies suggest that high cholesterol levels may also significantly contribute to COPD progression or other comorbidities. A clinical randomized cross-sectional study enrolling 100 patients with severe and very severe COPD demonstrated that patients with very severe COPD had a positive correlation with low-density lipoprotein (LDL)-C (r = 0.895, *p* < 0.001) and had elevated levels of cholesterol (6.16 ± 1.5 vs. 5.61 ± 1.1, *p* = 0.039) [4]. Another recent multi-center COSYCONET (COPD and SYstemic consequences-COmorbidities NETwork) cohort included 1746 COPD (Gold class 1-4) patients who demonstrated that hyperlipidemia was associated with lower intrathoracic volume and higher forced expiratory volume in 1 s (FEV1) suggesting that COPD patients with hyperlipidemia showed less lung hyperinflation and airway obstruction [1]. Another cross-sectional observational study with 125 different classes of COPD severity patients found that hyperlipidemia was common in combination with COPD, but no statistical difference was observed for total cholesterol (TC), LDL, very low-density lipoprotein (VLDL), and triglycerides between mild, moderate, and severe COPD [5]. A cross-sectional study investigating 644 adolescents aged 15–16 years old showed that the percent predicted values of forced vital capacity (FVC) (*p*  =  0.036) and FEV1 (*p*  =  0.017) were also inversely associated with high-density lipoprotein (HDL) cholesterol [6]. HDL-cholesterol levels also correlate with muscle areas (pectoralis, rectus abdominis, and erector spinae), densities, and total energy expenditure in patients with or at risk for COPD [7]. Recently, several studies suggested a relationship between body mass index (BMI) and COPD. Some studies observed a negative impact of BMI on COPD [8,9], while others suggest a better prognosis with excess body weight [10]. This “obesity paradox” is a topic for discussion, and cholesterol could play a significant role, but additional research is needed on this lifestyle topic [11,12]. Patients with COPD are markedly inactive because of airflow limitation and shortness of breath [13]. Decreased active time correlates with cardiac event risk, mortality, and higher HDL cholesterol levels [14,15].

Smoking is believed to cause intracellular cholesterol accumulation that plays an important role in lung inflammation in COPD [16]. Cholesterol is also a vital component of surfactant, a complex mixture of lipids and proteins secreted by alveolar type II epithelial cells that helps reduce the work of breathing by lowering the surface tension during inspiration [17]. However, dysregulation of cholesterol homeostasis and overloading macrophages with cholesterol in COPD plays cytotoxic and inflammatory responses. Advanced COPD is associated with the generation of inducible bronchus-associated lymphoid tissue (iBALT). Oxysterols, the metabolites of cholesterol, are elevated in COPD patients and are involved in iBALT generation and immune pathogenesis [18]. Therefore, several studies have investigated the potential benefit of cholesterol-lowering medications in COPD patients. 3-hydroxy-3-methyl-3-glutaryl coenzyme A reductase inhibitors (statins), reduce the expression of chemokines, such as CCL2 and CXCL8, and matrix metalloprotease (MMP) 9, all of which are increased in COPD [19]. A recent meta-analysis that included eight studies and 1323 participants showed that statins reduced cholesterol and lung inflammation without any clear improvement in functional capacity or lung function [20]. Another network meta-analysis study suggested that statins reduced the risk of all-cause and cause-specific mortality, pulmonary hypertension, and the level of C-reactive protein (CRP), IL6, IL8, and TNF-α in COPD patients [21]. However, statins may not impact exacerbation rates in patients with COPD [22], which is an important factor in COPD progression. In animal models of COPD, inhibition of the oxysterol pathway with the CYP7B1 inhibitor clotrimazole resolved B cell-driven iBALT formation and attenuated cigarette smoke-induced emphysema [23]. Here, we will briefly discuss the role of smoke exposure on cholesterol levels and responses in COPD and review the literature on the subsequent impact of cholesterol on inflammation and mitochondrial function within the lungs.

## 2. Is There a Relationship between Smoking Status and Lung Cholesterol Levels?

Several studies investigated blood or bronchoalveolar lavage (BAL) fluid cholesterol levels and whether smoking or disease status are associated with these changes. The effects of smoking on serum lipids suggest that it induces an atherogenic lipid profile which in turn has ramifications for cardiovascular well-being as well as systemic inflammation. A one-year, prospective, double-blind, randomized, placebo-controlled clinical trial demonstrated that smoke cessation leads to increases in HDL cholesterol, total HDL, and large HDL compared to individuals that continue to smoke, especially in women [24]. There were no significant changes in LDL cholesterol with smoke cessation [24]. In current smokers, increased smoking inhalation is associated with increases in total cholesterol, LDL cholesterol, and triglycerides [25]. CO levels were also associated with lower HDL-cholesterol and total HDL particles [25]. Even with second-hand smoke, HDL cholesterol levels are lower in dyslipidemic children [26]. A recent publication demonstrated elevated plasma levels of cholesterol in COPD patients [27]. The exact mechanisms for how smoking affects the lipid profile are not understood but are postulated to be related to catecholamine release with subsequent changes in circulating free fatty acids and resultant increases in VLDL and LDL with a decrease in HDL [28].

Total lipid concentrations are decreased in the BAL fluids of COPD patients compared to healthy controls, and cholesterol is the second most abundant lipid in the airways [29]. In COPD patients, the amount of cholesterol in BAL fluid is reduced compared to healthy controls, which is also positively correlated [29]. This loss of BAL fluid cholesterol is also observed in smoke-exposed mice [29]. A recent preclinical study suggested that the foamy macrophage phenotype and inflammatory response (primarily IL-1α) following cigarette smoke exposure may be due to the direct damage made to pulmonary lipids [30]. The investigators also observed that intranasal delivery of human oxidized low-density lipoprotein led to a similar macrophage response to cigarette smoke exposure [30], which was attributed to altered surfactant responses.

Cigarette smoke can impair the surface tension lowering the ability of surfactants. The lipid constituents in surfactant aid in their ability to reduce surface tension, facilitate gas exchange, and help in surfactant production and immune modulation [31]. A systematic lipidomic analysis of BAL fluid in subjects with COPD that had achieved smoking cessation was compared with BAL from healthy subjects, and observed a decrease in alveolar lipids, which correlated with a reduction in lung function [29]. Smoking decreases BAL fluid lipids, especially phospholipids and cholesterol, followed by sphingolipids and glycerolipids [32]. Decreases in total BAL lipid, total PL, PC 30:0, PC 32:0, and total cholesterol, amongst other lipids, strongly correlated with decreased lung function, which was also observed in the second-hand smoke mice model [29]. Multiple mechanisms are postulated for the effect smoking has on pulmonary lipids, including the destruction of type II pneumocytes, decreased synthesis, and alterations in transport [33,34]. It is important to note that a small clinical trial showed improved lung function in patients with stable chronic bronchitis who were given supplemental surfactant therapy [35]. This further suggests that pulmonary lipids/surfactants may play a role in the pathophysiology of COPD.

## 3. Cholesterol Levels Can Influence Systemic and Local Lung Inflammation in COPD

Hypercholesterolemia leads to cholesterol accumulation in many cell types, especially immune cells, that contributes to elevated inflammatory responses, including inflammasome activation, Toll-like receptor (TLR) signaling, and the production of innate immune cells [36]. Elevated levels of LDL promote cholesterol accumulation and heightened inflammatory response. To opposes this, HDL promotes the cellular efflux of cholesterol and reduces inflammation. Since HDL can suppress inflammation, some investigators have suggested that it plays a protective effect on the smoke-exposed lung [37]. Decreased HDL could contribute to more inflammation within the lung. HDL stimulates the release of cholesterol from activated cholesterol-filled macrophages, which diminishes their inflammatory response in mice [38]. In acute septic patients and septic animal models, cholesterol efflux from macrophages to plasma or HDL enhances inflammation [39].

In macrophages, cigarette smoke impairs cholesterol efflux by reducing the expression of ATP binding cassette (ABC) A1 leading to enhanced TLR4/myeloid differentiation primary response 88 (MyD88) signaling, MMP9, MMP13, TNFα, IL1β, IL17, IFNγ expression, and suppression of IL10 that are also responsive to liver X receptor (LXR) agonists in mice [34]. The same study demonstrated that treatment with an LXR agonist, T0901317, prevented cigarette smoke-induced loss of lung function in AKR/J mice [34]. Cigarette smoke directly impacts cholesterol efflux in macrophages and reduces the frequency of foamy cell-like/lipid-laden macrophages in the lungs [34]. Another LXR agonist, GW3965, suppressed the production of lipopolysaccharide (LPS)-dependent CXC10 and CCL5 and elevated IL-10 production in alveolar macrophages [40]. A study conducted in *Abca1* deficient mice demonstrated reduced airway eosinophils, T helper (Th) 2 cytokines, and Th2 cells, and increased airway neutrophils and IL-17 [41]. Therefore, ABCA1 plays a crucial role in cholesterol-mediated lung inflammation in cigarette smoke exposures. A recent transcriptome and lipidome study focusing on macrophages from COPD patients demonstrated that there are cholesterol metabolism and IFN-α and γ signatures within the macrophages that are GOLD grade-dependent [42]. In mice, cholesterol efflux pathways mediated by apolipoprotein (APO) E, ABCA1, and ABCG1 suppress the production of inflammatory cells in the bone marrow and the spleen [43]. This is also relevant to humans, as there is an inverse relationship between HDL cholesterol levels and blood monocyte numbers in children with familial hypercholesterolemia [44]. Therefore, elevated cholesterol influences immune cell proliferation in addition to inflammation responses. Promoting reverse lipid transport with recombinant ApoA-1 Milano/phospholipid complex (MDCO-216) prevented smoke-induced immune cell infiltration and partially protected lung function in mice [33].

Recently, Li et al. demonstrated that immune responses in airway epithelial cells are also influenced by cholesterol [27]. They observed that cigarette smoke extract (CSE)-induced expression of IL8 and IL6 in human bronchial epithelial (HBE) cells could be reversed by either a statin (atorvastatin) or LXR agonists (T0901317 and GW3965) [27]. Equally, they identified that the steroidogenic acute regulatory-related lipid transfer domain-3 (STARD3) was a critical player in cholesterol and cigarette smoke-induced IL6, IL8, and IL1β [27]. STARD3 is a sterol-binding protein that creates endoplasmic reticulum (ER)-endosome contact sites, and STARD3 can regulate cholesterol accumulation in endosomes rather than the plasma membrane [45]. Cholesterol regulation of STARD3 expression may be a mechanism for how smoke exposure modulates epithelial inflammation response sensitivity within the lung, and it also appears to influence cholesterol levels in the mitochondria. It may contribute to mitochondrial dysfunction [27]. STARD3 also significantly increases macrophage ABCA1 expression, enhances cholesterol efflux to APOA1, and reduces biosynthesis of cholesterol, cholesteryl ester, fatty acids, triacylglycerol, and phospholipids [46]. The overexpression of STARD3 (also known as MLN64) in osteoclasts enhances RANKL induction of inflammatory factors, including NFκB, TNFα, IL1β, IL6, and MMP1 [47]. Additionally, regulating plasma membrane cholesterol content may be necessary for regulating neutrophil-associated inflammation and adhesion, as maintaining cholesterol membrane content coincides with reduced inflammation in other lung diseases, such as cystic fibrosis [48].

The impact of elevated cholesterol on lung surfactants must also be considered as surfactant composition plays a major role in lung surface tension, but surfactant also has immune properties, with extracellular large surfactant aggregates observed to bind to non-typeable *Haemophilus influenzae* to inhibit adhesion of the bacterium to pneumocytes and prevent invasion [49]. Furthermore, in rodents, over 80% of the cholesterol in the rat lung comes from cholesterol uptake from circulation [50]. Therefore, elevated systemic cholesterol will influence cholesterol uptake levels in the lungs and could impact surfactant composition and function.

Finally, the impact of hypercholesterolemia on TLR signaling and 25-hydroxycholesterol (25HC), a metabolite of cholesterol that is produced and secreted by macrophages in response to TLR activation [51], is extensively studied in atherosclerosis. However, 25HC is also known to be elevated in COPD lungs, and its levels inversely correlate with important readouts of lung function, such as the percent of predicted FVC, FEV1, and diffusing capacity of carbon monoxide (DLCO) [52]. 25HC is encoded by the *Ch25h* gene, a known IFN-inducible gene. Many inflammation triggers, such as LPS, polyinosinic-polycytidylic acid (poly(I:C)), viruses, and interferons, can upregulate *Ch25h* expression and increase the production and release of 25HC in macrophages [53]. 25HC can also inhibit the growth of a broad range of viruses. Recently, 25HC inhibited SARS-CoV-2 and other coronaviruses by depleting membrane cholesterol [54]. Therefore, intra- and extra-cellular cholesterol, cholesterol efflux, and metabolites of cholesterol will affect immune responses in COPD and influence how the lung responds to inflammatory stimuli.

## 4. Hypercholesterolemia and Mitochondrial Dysfunction

The mitochondria play critical roles in energy production via oxidative phosphorylation, cell signaling, and maintaining tissue homeostasis. The mitochondria are regulated by a balance of mitochondrial membrane potential, mitochondrial calcium, mitochondrial DNA (mtDNA), and clearance of impaired mitochondria via mitophagy. However, many different insults may disrupt mitochondria regulation within the lung, including inflammation, infection, air pollution, cigarette smoke, and alteration of oxygen levels. These insults may cause mitochondria dysfunction via multiple different mechanisms, including mtDNA impairment and release, increased reactive oxygen species (ROS), reduced mitochondrial membrane potential, and dysregulation of calcium signaling and storage [55]. These alterations can ultimately induce metabolic reprogramming, inflammasome activation, apoptosis, senescence, oxidative stress, and inflammasome activation [56]. Recently, hypercholesteremia was demonstrated to induce mitochondria dysfunction in airway epithelial cells in COPD [27].

In preclinical models of the disease, hypercholesteremia is observed to induce mitochondrial dysfunction [57]. Feeding C57B1/6J mice a high-fat diet resulted in the downregulation of genes in mitochondrial oxidative phosphorylation [57]. This was further elucidated in humans where a high consumption of fat diet-induced dysregulation of target genes involved oxidative phosphorylation. Another potential mechanism identified is the accumulation of cholesterol within the mitochondria, causing a decrease in the mitochondrial glutathione stores, a key component in ATP production, by preventing its transport into the mitochondria [57]. Multiple studies have demonstrated that the accumulation of fatty acids within the inner mitochondrial membrane increases susceptibility to peroxidation leading to mitochondrial dysfunction [58]. The enzyme carnitine palmitoyltransferase (CPT) 1, which allows for the inward transportation of fatty acids, is linked to these fatty acids-associated mitochondrial dysfunction. Another method is through passive diffusion, described as the ‘flip-flop’ mechanism [59]. It explains how cholesterol bilayer motion influences cholesterol distribution within the cell membrane. Investigating the mitochondria genome may provide insights into understanding the physiological mechanisms by which lipid levels affect mitochondrial function and the possible influence of mtDNA on lipid changes [60]. One study identified 10 mitochondrial single nucleotide polymorphisms (mtSNPs) associated with HDL and cholesterol [60]. While there are many studies conducted to investigate the mechanisms by which hypercholesterolemia affects mitochondria function, further research is needed to identify these changes in the immunopathology of various diseases.

Mitochondrial dysfunction plays a critical role in driving the inflammatory process of lung inflammation in patients with COPD [61]. HBE cells (the cell line BEAS-2B) exposed to prolonged CSE or primary HBE cells isolated from COPD subjects have abnormal mitochondria, observed with increased mitochondrial mass, fragmentation, branching, and quality of cristae [62]. Reduced expression of the peroxisome proliferator-activated receptor-gamma coactivator (PGC) 1α, a key regulator of mitochondrial biogenesis regulated by sirtuin-1, is linked to the dysregulation of mitophagy, a cellular process to selectively remove aged and damaged mitochondria and cellular senescence in the lung [63]. Increased mitochondrial iron is also linked to mitochondrial dysfunction in COPD patients [64]. Mitochondrial dysfunction in macrophages from COPD patients is also linked to defective bacterial phagocytosis [65]. Extracellular mtDNA can activate the NLRP3 inflammasome [66] and the cytosolic DNA sensor cyclic guanosine monophosphate-adenosine monophosphate synthase [67]. Both of these responses would lead to the activation of innate immunity and the secretion of inflammation mediators. Increased mtDNA is observed in the urine of COPD patients and correlates with disease severity [68]. Mitochondrial dysfunction may play a key role in the pathogenesis and progression of COPD, and hypercholesterolemia may be a confounding factor for mitochondrial dysfunction in COPD.

There are large aggregates of mitochondria within muscle fibers of the diaphragm of a COPD patient [69] that may cause an increase in cytochrome oxidase activity and ROS production. The known mechanisms by which the mitochondria induce inflammation in COPD patients are a decrease in oxidative capacity, increased ROS production, and increased apoptosis and autophagy. The signaling pathways of iron-regulatory proteins, prohibitin complexes, PPAR-y coactivator, and peroxisome proliferator-activated receptors have also been demonstrated to contribute to lung inflammation in COPD patients. Interestingly, there is growing evidence that hypercholesteremia may significantly contribute to mitochondria dysfunction in COPD patients [27]. Preclinical studies suggest that there is an increase in cholesterol biosynthesis in lung epithelial and lipofibroblasts during aging, indicating a potential link between hypercholesteremia and HBE function in driving inflammation in disease [70]. A recent study by Li et al. in 2022 demonstrated that the external cholesterol level alterations influenced mitochondrial dynamics leading to a dysregulation of interleukin production via the down-regulation of the SREPB2/LDL-R pathway [27]. In addition, the STARD cholesterol transporters are key players in cholesterol distribution within and outside the cell and inflammation mediators, as mentioned in Section 3. The STARD family includes 15 different members. Of the 15 members, STARD1, STARD3, STARD4, and STARD5 regulate cholesterol homeostasis and assist in cholesterol transport. STARD3 can facilitate the transport of cholesterol intracellularly from endosomes to the mitochondria [45]. This may ultimately lead to cholesterol accumulation in the mitochondria resulting in increased oxidative stress and enhancement of steroidogenesis [45]. STARD3-mitochondrial fusion genes, including mitofusin (MFN) 2, may contribute to the induction of mitochondria dysfunction leading to the development of COPD [27].

Moreover, this study demonstrated that hypercholesteremia upregulates the expression of STARD3, which led to increased sensitivity of HBE cells to smoking and altered interleukin and cytokine production [27]. The authors suggested that there may be a dependency of STARD3 on airway epithelial sensitivity. When STARD3 is downregulated in preclinical models, it reversed the epithelial mitochondrial function and decreased sensitivity to various insults [27]. In addition, alterations in external cholesterol levels can directly influence mitochondria function [27]. It can downregulate inflammatory pathways, including the SREBP2/LDL-R pathway, and alters the transcriptomic expression of mitochondria-associated genes [27].

During cigarette smoke exposure, MFN2 expression levels were increased in human airway epithelia, while deletion of MFN2 altered the production of lipids [27]. MFN2 enhancement may be a compensatory response to cigarette smoke exposure. Chronic cigarette smoke exposure altered the STARD3 levels, which also altered the cholesterol levels within the mitochondria and externally [27]. This induces mitochondrial dysregulation leading to the accumulation of cholesterol, damaging its architecture and inducing activation of inflammatory signaling cascades. These findings demonstrated by Li provide new insights into how COPD patients with hypercholesteremia develop the disease, identify potential therapeutic targets, and possible indicators of disease severity. While significant progress has been made to identify the mechanisms by which hypercholesteremia induces mitochondria dysfunction leading to the development of disease, further research is needed to unravel new potential therapeutic targets and indicators of the disease process (See Figure 1 for possible mechanisms to cholesterol-induced inflammation and mitochondrial dysfunction in COPD).

## 5. Conclusions

Smoke’s regulation of cholesterol transport, inflammation, and mitochondrial dysfunction appears to be mechanistically linked to each other and may play a significant role in the increased risk of COPD and targeting cholesterol responses may represent a novel therapeutic approach. Epidemiologic studies suggest a link between pulmonary lipids and changes in pulmonary function from cigarette smoke exposure. However, only a few studies have identified pathogenic links of cholesterol to COPD pathogenesis. The effect of cigarette smoke on the expression and activity of the major ABC transporters, STARD3, MFN2, inflammation sensitivity, and mitochondrial dysfunction within the lung suggests that the disruption of lipid homeostasis is an early event in the development of COPD after cigarette smoke exposure. Here, we primarily focused on cholesterol changes, but other lipids and mediators are linked to COPD progression and also need to be discussed. Effects of race, gender, and diet are also confounding factors.

Similarly, we did not discuss the effects of cigarette smoking on cholesterol metabolism, biosynthesis, maturation, remodeling of cholesterol, cholesterol subfractions, and cholesterol catabolism. It will also be important to investigate the effect of cigarette smoke-induced oxidative modifications on cholesterol, lipid peroxidation, and other induced dysfunctions. Genetic factors, such as alpha-1 antitrypsin deficiency, are also major factors in COPD development and progression. Alpha-1 antitrypsin augmentation therapy is reported to coincide with reduced cholesterol levels in an observational study [71], but further research is needed. Furthermore, new cholesterol-targeting therapies must be extensively investigated in COPD subjects, such as proprotein convertase subtilisin/kexin type 9 (PCSK9) inhibitors and inclisiran, a small interfering RNA therapy. Recent studies suggest that metabolism does not progress as previously thought, with organ-specific metabolic activity remaining high throughout growth and development from infancy to adolescence. However, metabolism declines throughout adulthood and old age [72]. This may have many outcomes on our view of aging and COPD progression. Finally, there is evidence that vaping products, such as electronic cigarettes, can yield a similar lipid profile to conventional tobacco smoking [73], which could have many long-term complications. Therefore, significant work is required to elucidate the causality of altered lung cholesterol metabolism and signaling within the lungs, but it represents an important and exciting study area for pulmonary diseases such as COPD.

## Figures and Tables

**Figure 1 medicina-59-00253-f001:**
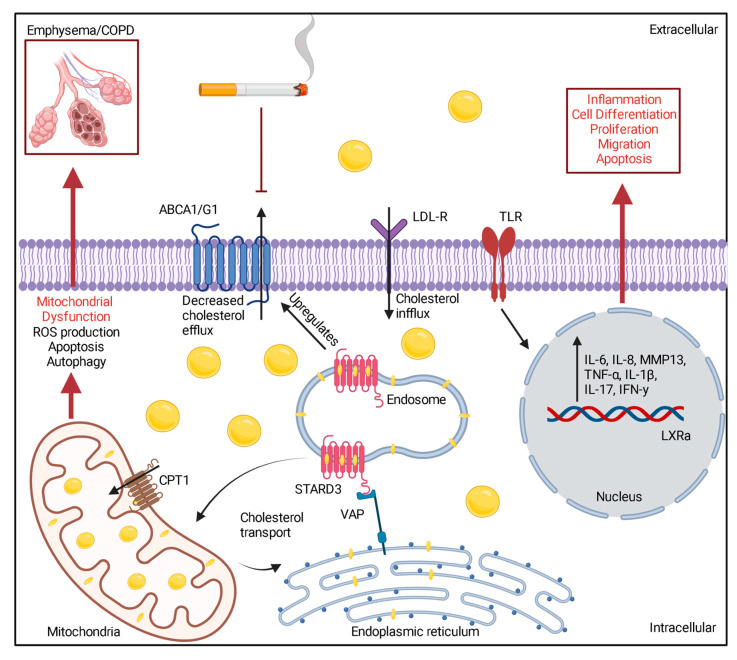
Possible cholesterol signaling that contributes to mitochondrial dysfunction, inflammation, and loss of lung function in COPD. Created with BioRender.com.

## Data Availability

Not applicable.

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
