# Peer review of "The Relationship of Cholesterol Responses to Mitochondrial Dysfunction and Lung Inflammation in Chronic Obstructive Pulmonary Disease"

_medicina, 2023, doi:10.3390/medicina59020253_

Round 1
Reviewer 1 Report
Hello,
Thank you for submitting the manuscript titled “Cholesterol responses linked to mitochondrial dysfunction and lung inflammation in COPD”. Authors have summarized the literature well and presented with cellular changes with figure for mitochondrial dysfunction in COPD. There are further points that authors should consider elucidating the correlation of higher cholesterol accumulation and COPD.
Minor:
1. As one would expect the lifestyle factor apart from genetic in progression of cholesterol accumulation. Authors should elaborate this area with related literature in COPD, discuss how lifestyle can impact the high cholesterol and worsening the COPD level.
2. Authors should discuss the pharmacological intervention and success so far in this domain.
3. Ageing factor can also change the progression of COPD and also metabolism in aged patients vs young patients can be lower in process of cholesterol. Authors should also add some related literature in this area with latest trend.
Thank you
Author Response
Reviewer 1:
- As one would expect the lifestyle factor apart from genetic in progression of cholesterol accumulation. Authors should elaborate this area with related literature in COPD, discuss how lifestyle can impact the high cholesterol and worsening the COPD level.
Response: We have added several sentences to address this topic. Please see lines 51-58
- Authors should discuss the pharmacological intervention and success so far in this domain.
Response: We added lines 78-80, and 334-336
- Ageing factor can also change the progression of COPD and also metabolism in aged patients vs young patients can be lower in process of cholesterol. Authors should also add some related literature in this area with latest trend.
Response: We have the lines 336-340
Reviewer 2 Report
I would suggest to change the title to make it more clear. The title of the paragraph no.4 and 5 is almost the same. Please change and reorganise. Please highlight and group the animal studies and clinical studies in the sections. Although smoking is the main cause of the COPD, there are also other causes of this disease. The authors should specify and focus on certain pathophysiological links and mechanisms.
Line 50: „this mechanism”
Unclear - Please express more precisely Line 98: „they also observed” Unclear - Please specify Line 102: unclear - please correct Line 126 to colloquial - please correct Line 148 this is also relevant to humans - please expres more precisely Line 159 - atorvastatin is also a statin - please correct Line 188 „in diseases like” to colloquial - please correct Line 313 „destruction of the lung” - too general, please correct Please distinguish the limitations of the study sectionAuthor Response
Reviewer 2:
I would suggest to change the title to make it more clear.
Response: We have modified the title
The title of the paragraph no.4 and 5 is almost the same. Please change and reorganise.
Response: We have removed the title for section 5 and combined both sections.
Please highlight and group the animal studies and clinical studies in the sections.
Response: All animal studies are now clearly labelled.
Although smoking is the main cause of the COPD, there are also other causes of this disease. The authors should specify and focus on certain pathophysiological links and mechanisms.
Response: We have added several sentences on alpha-1 antitrypsin deficiency. Please see lines 30-31, and 332-334
Line 50: „this mechanism” Unclear - Please express more precisely Line 98: „they also observed” Unclear - Please specify Line 102: unclear - please correct Line 126 to colloquial - please correct Line 148 this is also relevant to humans - please expres more precisely Line 159 - atorvastatin is also a statin - please correct Line 188 „in diseases like” to colloquial - please correct Line 313 „destruction of the lung” - too general, please correct Please distinguish the limitations of the study section
Response: We have reorganized these sentences to be clearer
Round 2
Reviewer 2 Report
The authors made the sufficient corrections to the manuscript. Please add inclisiran as a novel therapeutic option in line 336
Author Response
We thank the reviewer for their helpful comments. We have added inclisiran as a novel therapeutic option on line 336